# Layered liquid crystal elastomer actuators

Tyler Guin[1,2], Michael J. Settle[3,4], Benjamin A. Kowalski[1,2], Anesia D. Auguste[1], Richard V. Beblo[3,4], Gregory W. Reich[3] & Timothy J. White[1]

Liquid crystalline elastomers (LCEs) are soft, anisotropic materials that exhibit large shape transformations when subjected to various stimuli. Here we demonstrate a facile approach to enhance the out-of-plane work capacity of these materials by an order of magnitude, to nearly 20 J/kg. The enhancement in force output is enabled by the development of a room temperature polymerizable composition used both to prepare individual films, organized via directed self-assembly to retain arrays of topological defect profiles, as well as act as an adhesive to combine the LCE layers. The material actuator is shown to displace a load >2500× heavier than its own weight nearly 0.5 mm.

[1] Air Force Research Laboratory, Materials and Manufacturing Directorate, Wright-Patterson Air Force Base, OH 45433, USA. [2] Azimuth Corporation, 4027 Colonel Glenn Hwy, Beavercreek, OH 45431, USA. [3] Air Force Research Laboratory, Aerospace Systems Directorate, Wright-Patterson Air Force Base, OH 45433, USA. [4] University of Dayton Research Institute, 1700 S Patterson Blvd, Dayton, OH 45469, USA. Correspondence and requests for materials should be addressed to T.J.W. (email: timothy.white.24@us.af.mil)

Responsive materials are currently subject to intense research, motivated in part by end-use applications in robotics[1]. An emerging research frontier is materials systems that inherently emulate the motion, dexterity, and force output of natural musculo-skeletal systems[2]. An increasingly common approach is to locally vary the organization or other materials properties such that "the material is the machine[3]." Realization of real world devices will require further innovation and development of both hard and soft materials.

In some implementations, shape reconfigurability will be an important aspect of robotic control. Stimuli-responsive shape change of monolithic elements is exhibited by a range of material platforms, including shape memory alloys (SMAs)[4]. SMAs achieve large force output but limited deformation, and are found in end-use applications in medicine, automobiles, and aerospace[5]. Recent explorations focus on soft materials in which the mechanical response can be localized and potentially programmed, at the expense of output force[6].

Natural musculo-skeletal systems employ anisotropy to optimize function, as well as grade the interfacial interaction of stiff and soft elements. Liquid crystalline elastomers[7] have been studied for nearly 50 years, traceable to original predictions from de Gennes[8]. Uniquely, the molecular orientation of these materials can be programmed pixel-wise with micron-scale resolution. Robust and high-throughput patterning is enabled by exploiting directed self-assembly (both spatial and hierarchical) onto a patterned template surface[9]. The molecular orientation governs the anisotropy of macroscopic mechanical response, and monolithic elements composed of these materials can be permanently programmed to exhibit reversible, stimuli-responsive shape transformations. A wide range of shapes can be realized such as origami folds[9–11], arrays of cones[12], or arbitrary curvatures, such as paraboloids[13]. Notably, these materials are continuous in composition and absent of multimaterial interfaces. Mechanical responses in these materials can be triggered by exposure to heat[14, 15], light[16, 17], electrical fields[18, 19].

The tremendous shape transformation of LCEs can create useful work. LCEs with uniform orientation (via mechanical stretching) exert muscle-like contractile force generating strains of up to 400%[20]. A number of recent reports detail a comparatively distinctive approach to generating force. LCE sheets with spatially patterned orientation can act as out-of-plane lifters, using shape change to generate considerable work over a large stroke, with a work capacity of as much as 2.6 J/kg[9] (from a soft material of 50 μm thickness). The extraordinary work capacity of these materials is attributable to the fundamentals of the shape transformation. The spatial variation in the director profile dictates that the material must emanate into a third dimension, via stretch. It is predicted in ref. [21] that force outputs should correlate to increasing the film thickness. However, the achievable thickness of LCEs prepared by surface anchoring is limited. For cell thicknesses exceeding roughly 50 μm, the patterned alignment surface can no longer effectively prescribe alignment through the entire cell, due to finite anchoring energies of surface interactions[22].

Here we present a novel method to create arbitrarily thick LCE films that are continuous in composition and maintain complex director orientations, prescribed into the material via directed self-assembly by photoalignment. Our approach is to laminate as many as six LCE films bonded with interfacial layers of the same composition. Critically, these adhesive layers take on the residual orientation of the adjacent LCE layers. To enable this process, we develop a LCE composition with a room temperature nematic phase. The laminates maintain the shape transformation of single layer LCE films when heated. However, the laminated films exhibit extraordinary lifting forces to nearly 20 J/kg (as much as

2500× the weight of the mechanical system). We illustrate that these materials are now capable of withstanding significant positive pressure, which could open up end uses in aerospace and other application domains.

## Results and discussion

**Materials preparation and characterization.** The LCE films examined here were formulated by mixing mesogenic diacrylates (RM82 and RM257) with a dithiol chain-transfer agent (Fig. 1a). As detailed in [23], the dithiol additive reduces the crosslink density of the polyacrylate via chain transfer (primary) and chain extension (secondary). The concentration of RM82 and RM257 in the mixture was selected to suppress the nematic-crystallization phase transition[24], producing a supercooled mixture which was meta-stable (>1 h) to −20 °C. The broad phase range enabled processing and photopolymerization to occur at room temperature. The LCE films are optically clear (Fig. 1b). The glass transition temperature ($T_g$) of the LCEs prepared from this composition was 26 °C, (Supplementary Figure 1) similar to prior reports[13]. Local organization of the monomeric mixture was directed by photoalignment cells (PAAD-22, BEAM Co.)[25]. Here we impose various director profiles into the material to localize the orientation of the liquid crystalline monomers into topological defects, which are subsequently retained after photopolymerization.

Dating to de Gennes' original predictions[8], LCEs have been discussed as synthetic candidates to emulate the tremendous force output of muscle fiber[15]. Examinations of LCEs in a planar (i.e., monodomain, single crystal) orientation have reported force outputs of 250 kPa[15]. Recent examinations of LCEs imprinted with complex topologies, such as the examination here, have reported specific work capacities of 2.6 J/kg[9]. The focus of this work is to explore and demonstrate approaches and considerations to substantially increase the work capacity of LCEs while maintaining (or extending) the large stroke.

An approach to increase the work capacity of these material systems is to increase the thickness of the LCEs[26]. However, the physics of surface anchoring[27] and the anchoring strength of the photoalignment layer[28] limit the maximum thickness for retention of surface-induced director profiles to ~50 μm[22]. To circumvent, we report here a newly developed method to laminate LCE films (illustrated in Supplementary Figure 2). After polymerization and under crossed polarizers, two LCE films of identical composition and director profile were registered. Leveraging the room temperature nematic phase of the monomer composition employed to prepare the films, the samples were coated with a thin layer of the monomer mixture (same composition as above). After registry, the two LCEs were sandwiched together and gently clamped. The samples and monomer mixture were briefly heated and slowly cooled, to allow the adhesive layer to take on the residual surface alignment of the films. The films were cured by UV light exposure. From this two layer laminate, additional layers can be added to realize up to 300 μm thick LCE laminates examined here. Due to the consistency in the materials chemistry acting as both the LCE layer and adhesive, we observed no delamination in any of our experiments.

The orientation of the laminated films was confirmed with polarized microscopy[29] to confirm the registry of the LCE layers and that the adhesive layers are taking on the order of the LCE surfaces (Supplementary Figure 3). The thermally induced contraction (Fig. 1c) of the uniaxially aligned LCEs is nearly identical for a single layer, double layer, and four layer laminates. The contraction measured in Fig. 1c was determined from dimensional changes observed in the LCE films upon heating.

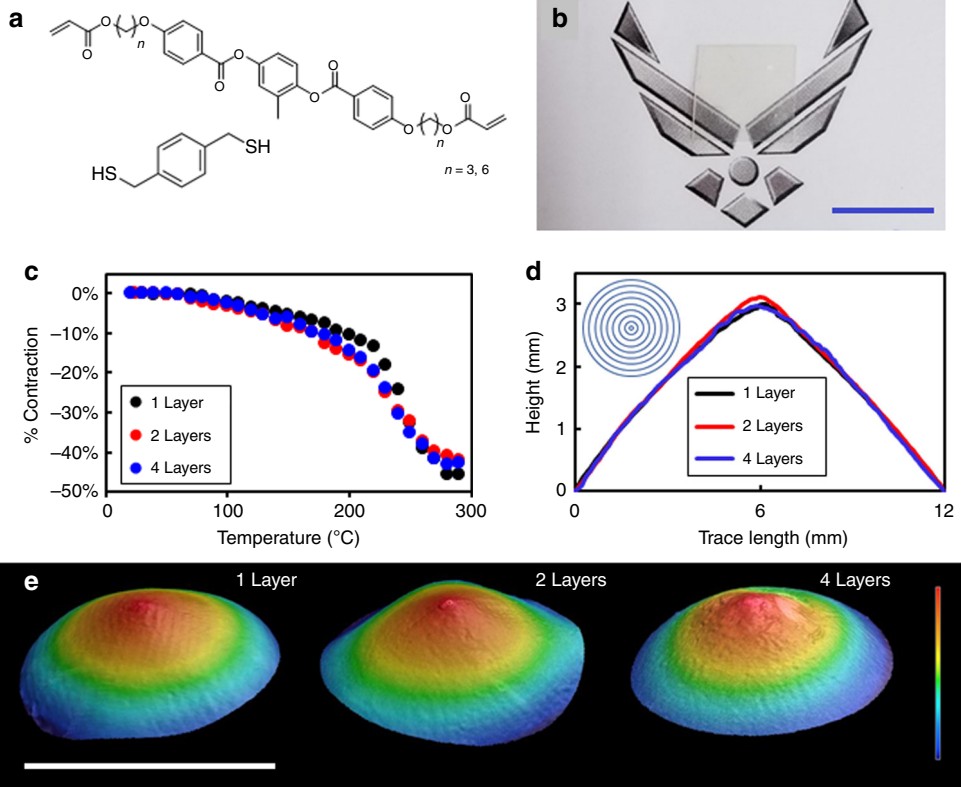

**Fig. 1** Unloaded deformation of LCE laminates. **a** Chemical structures of: (1) the liquid crystal monomers (RM257, RM82) and (2) the dithiol additive. A mixture of 7:2 wt:wt of RM257:RM82 formed the basis of the liquid crystal elastomers (LCEs). **b** A 4 layer LCE laminate is optically clear. Scale bar is 1 cm. **c** The deformation of one, two, and four layer LCE laminates was measured in homogeneously aligned LCE films. 50% strain is observed in all three samples. **d** The height profiles for the single layer, double layer, and four layer LCE laminates are nearly identical over the 12 mm diameter of the films. Horizontal scale bar is 12 mm, vertical scale bar is 3.4 mm. **e** Height profiles of LCE laminates in which the individual layers were subject to directed self-assembly to organize into radial +1 topological defects. The LCE laminates were heated to 140 °C and imaged with an optical profilometer to quantify the deformation to thermal stimuli

**Deformation of laminated LCEs**. When heated, nematic LCEs reversibly contract along the liquid crystalline director and expand in the orthogonal directions[30]. This anisotropic contraction, when subject to spatial variation dictated by the directed self-assembly of localized surface alignment, can result in dramatic out-of-plane shape deformation[14, 31]. Here we employ a well-understood and predictable director profile, the azimuthal +1 topological defect. The director profile is inset in Fig. 1d, where the mesogens organize in concentric rings around a central region (point defect). This pattern was predicted[31] and experimentally confirmed[14] to deform into a cone upon heating.

The deformation of single layer, double layer, and four layer LCE laminates was quantified by structured-illumination optical profilometry (Keyence VR-3000). The LCE film and laminates actuate into cones upon heating (Fig. 1d). The amplitude of the peak height (3.4 mm, ~70× the film thickness) and the angle of the cone tip are nearly identical among films of 50 μm (single layer), 102 μm (double layer), and 210 μm (four layer) thickness (Fig. 1e). The increased thickness of the laminates does not diminish the shape transformation. The insensitivity of the shape-morphing to film thickness is in agreement with a prediction of[32] that the deformation of an LCE sheet into a cone should be largely independent of sheet thickness, except for slight deviation near the tip. The agreement evident in Fig. 1d, e are strong, indirect evidence that each LCE layer as well as the adhesive interfaces are cooperatively deforming.

**Using the force**. In a previous report, a 2 × 2 array of +1 topological defects in a 50 μm thick LCE film was shown to lift up to 150× its weight with a stroke of 1 mm[9]. The stroke/force output of the single layer LCE films translates to a specific work capacity of 2.6 J/kg. Informed by the results in Fig. 1, we extend this examination to characterize and assess the potential actuation force of the LCE laminates. In Fig. 2, 1 × 1 cm LCE films patterned into a 2 × 2 arrays of radial +1 topological defects (director profile illustrated in Fig. 2a) were prepared. Single, double, and four layer LCE laminates were heated with a resistive heating element. A piece of glass was placed on top of the films, which was loaded with weight. As illustrated in the representative photograph in Fig. 2b, the deformation of the films is observable under load (28.7 g in Fig. 2b, see also Supplementary Movie 1). The deformation under load for one, two, and four layer LCE laminates are presented in Fig. 2c. Similar to the unloaded case in Fig. 1e, the stroke of the LCEs under load is relatively unaffected with increasing thickness. However, the increase in thickness increases the output force dramatically. The four layer LCE laminate (210 μm thick) produces 280 mN of force at a stroke of 1.6 mm. As illustrated in Fig. 2d, the specific work can reach nearly 19 J/kg. The four layer LCE laminate with the director configuration described in Fig. 2a can lift over 1100 times the weight of the film itself, a 100× improvement in specific work when compared to a single layer LCE.

When selecting actuators[33] both force output and stroke length are important considerations. So-called piezoelectric stacks can be

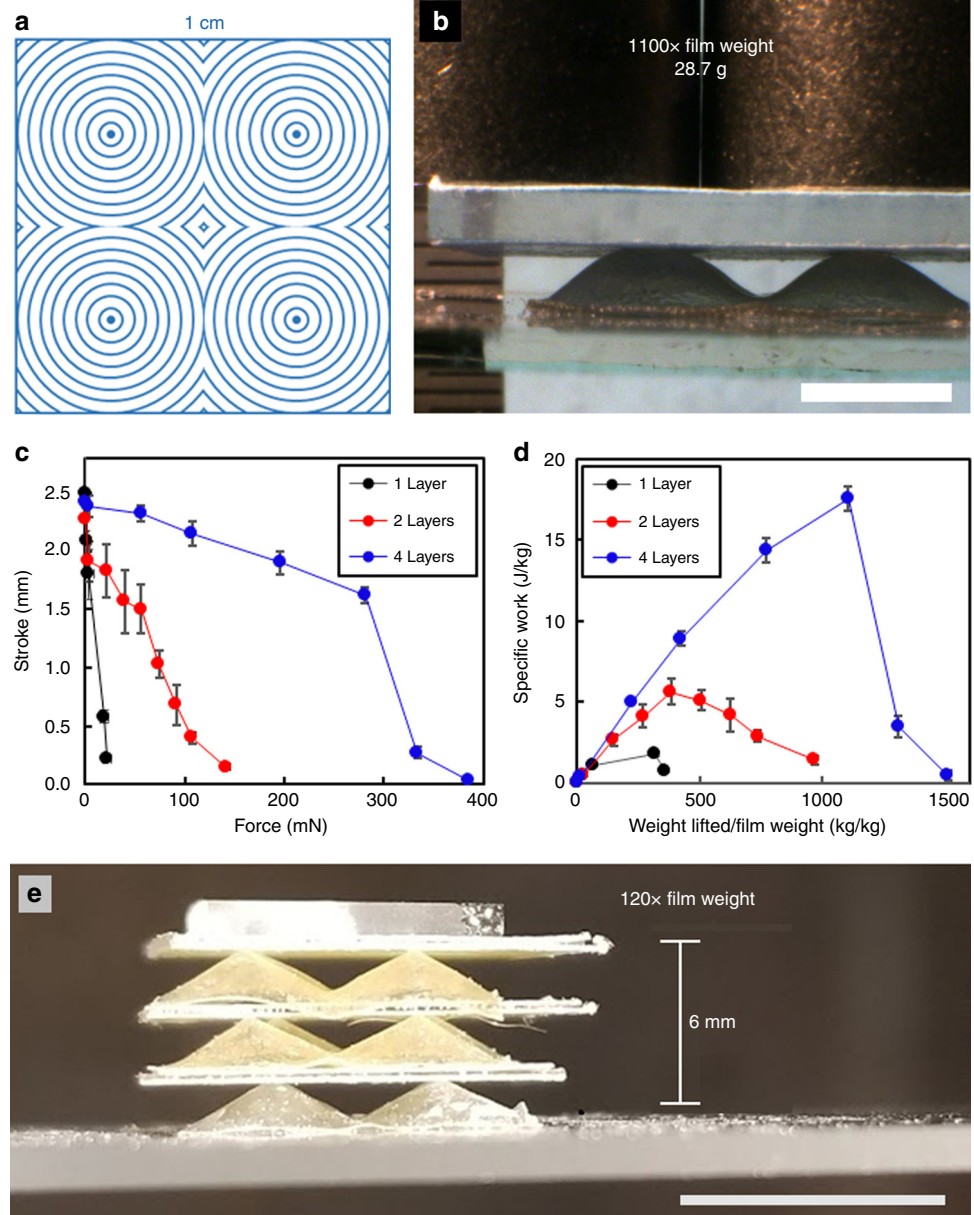

**Fig. 2** Soft weightlifting. **a** Prescribed director profile, to prepare a LCE films with a 2 × 2 array of radial +1 defects. **b** Deformation of a four layer LCE laminate lifting 28.7 g >2 mm. Scale bar is 3 mm. The LCE laminate was heated to 180 °C. **c** Stroke of LCE laminates under load. **d** The specific work of the LCE laminates is contrasted against normalized weight. Error bars represent statistical experimental error. **e** Three 1×1 cm LCE films are stacked with interfacing glass sheets, emulating a piezoelectric stack. A total of 1 g was lifted over 6 mm. Scale bar is 1 cm

emulated[34]. Evident in Fig. 2e, a large stroke actuator is demonstrated in which the films are stacked on top of each other, separated by a rigid substrate (glass coverslips). With three LCE layers, a stroke exceeding 6 mm is achieved while still lifting 120 times the weight of the assembled device.

A distinguishing characteristic of LCEs in contrast to other shape-changing polymeric systems is their excellent reversibility and resistance to fatigue[35]. Shape memory effects in polymers must be reprogrammed after each actuation[36, 37]. The robustness of the actuation of the LCE films examined here is illustrated in Fig. 3. Figure 3a, b contrasts the deformation of an LCE film under the load of nearly 1 g. There is little distinguishable difference in stroke or shape of LCE film after 10 thermal cycles. Figure 3c is representative of the flat state reached by the LCE films after each cycle. Figure 3d summarizes the consistency in the stroke observable in 11 thermal cycles.

**More is better**. We initiated this study by examining a 2 × 2 array of +1 topological defects in a 1 cm² film. The force output onto the loaded substrate in Fig. 2 should be sensitive to the number of contact points. To illustrate this, LCE films with a 3 × 3 array of +1 defects were prepared in 1 cm² films. By increasing the number of defects (contact points) from four to nine, the total force output increases from 300 (Fig. 2c) to 560 mN (Fig. 4a). Evident in Fig. 4b, the four layer LCE laminate composed of the 3 × 3 array of +1 defects is able to lift 2150 times the weight of the film itself. A four layer LCE film can lift 56 g nearly 0.5 mm (Fig. 4c).

Evident in Fig. 4a, the stroke is significantly decreased for the 3 × 3 arrays in the 1 cm² film when compared to the 2 × 2 arrays in the 1 cm² films examined in Fig. 2c. The direct comparison of 2 × 2 and 3 × 3 arrays in Fig. 4a, b are from samples in which the dimensions of each defect region was 0.33 × 0.33 cm[33]. These LCE

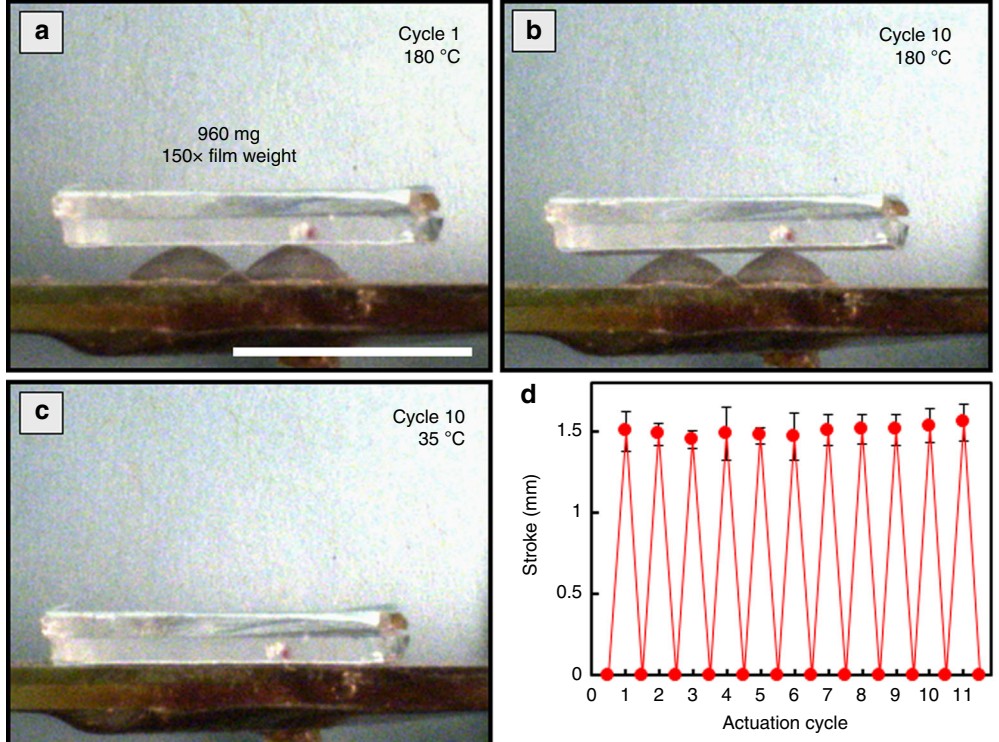

**Fig. 3** Actuation cycles. Four layer LCE laminate composed of layers with 2 × 2 array of +1 defects. Scale bar is 1 cm. Deformation of the four layer LCE laminate under 960 mg load was monitored **a** after 1 cycle, **b** 10 cycles (heating step), and **c** 10 cycles (cooling to 35 °C). **d** Stroke vs thermal cycle for 11 actuation cycles. Error bar is statistical accuracy of measurement

laminates exhibit identical stroke lengths. However, evident in Fig. 4a, the output force is considerably increased by increasing the number of contact points. Conceivably, employing large area patterning techniques and substrates not available to us in our laboratory could allow for preparing larger area films composed of 1000s of contact points.

**Deformation under pressure**. Numerous end-use applications of LCEs have been discussed including irises[38], biomimetic actuators[35], valves[39], and shape-changing lenses[40, 41]. One potential aerospace application is to prepare reconfigurable topographical surface features to manipulate flow[42]. We conclude this examination by measuring the deformation of the LCE laminates under positive pressure. LCE films were once again patterned with radial +1 topological defects. The deformation of a single defect subsumed in the center of a 12 mm diameter film was examined. The LCE laminates were placed in a pressure chamber where the back (or bottom) of the film was maintained at ambient pressure while the front (or top) is subjected to positive pressure. The entire chamber was heated and then the shape of the film is measured via optical profilometry. Figure 5 presents the optical scans of single, double, four, and six layer LCE laminates. Direct heating of the films in ambient pressure conditions results in the expected conical deformations (leftmost column of Fig. 5). Notably, the deformation of the films is less than that observed in free standing films, largely attributable to film anchoring. However, upon adding even slight positive pressure, the single layer film (50 μm) immediately loses its shape, compressing into the pressure chamber. The two layer LCE laminate behaves similarly. Informed by the prior results, the four layer LCE laminate (210 μm) withstands much higher loads and is able to maintain a cone-like shape at 1.5 kPa of pressure. A six layer LCE laminate (320 μm) is able to withstand over 7 kPa (>1 psi) and still

maintain a conical shape. Profiles of actuated samples are shown in Supplementary Figure 4 and Supplementary Figure 5.

Here we have demonstrated an approach to realize thick LCE films capable of large force output and stroke. Upon exposure to thermal stimulus, the LCE laminates deform into the expected shapes. Notably, despite the increase in film thickness in the LCE laminates, the deformations of the materials (e.g., the stroke) remain constant. The increase in thickness allow the laminates to impart work on objects >2000 times heavier than the laminates themselves. End-use applications in aerospace, such as reconfigurable topographical surfaces, require deformation to positive pressure. Six layer LCE laminates are shown to deform up to 7 kPa pressure and retain the expected conical deformation. In this way, this work is a critical step forward in opening up new opportunities for the implementation of these materials in applications ranging from aerospace, automobiles, and consumer goods.

## Methods

**Materials synthesis**. RM82 (1,4-Bis-[4-(6-acryloyloxyhexyloxy)benzoyloxy]-2-methylbenzene) and RM257 (1,4-Bis-[4-(3-acryloyloxypropyloxy)benzoyloxy]-2-methylbenzene) were purchased from the Synthon Chemicals, and recrystallized from methanol before use. LCE formulations were prepared by adding 69 wt% RM82, 20 wt% RM257, 11 wt% BDMT (benzenedimethanethiol, Sigma Aldrich), with 1 wt% Irgacure 651 (BASF), and 0.5 wt% butylated hydroxytoluene to a glass vial and thoroughly mixing. A total of 50 μm thick liquid crystal cells[9, 16] were filled via capillary action at 90 °C in the isotropic state, and then cooled slowly to 25 °C (5 min). The cells were then exposed to 365 nm UV (120 mW/cm$^2$) light for 20 min to initiate photopolymerization. After curing, the films were collected by soaking the cells in deionized water for 2 h. Care was taken to deconstruct the cells so the LCE films remain adhered onto one of the glass substrates, to prevent wrinkling. Two films, both adhered to a single glass substrate, were placed on a hot plate at 50 °C. A drop of the LCE formulation was placed on one of the films, and the films clamped together. The films were briefly heated using a 100 °C heat gun until the adhesive layer was no longer hazy, and then allowed to cool to 25 °C over 5 min. The adhesive was cured until 365 nm UV light for 20 min. This process was repeated until the desired number of layers was achieved.

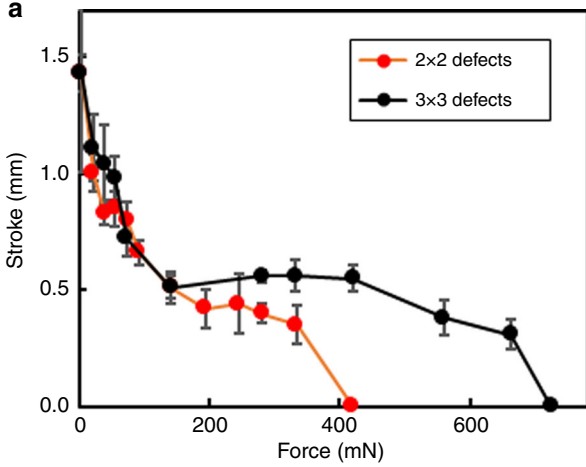

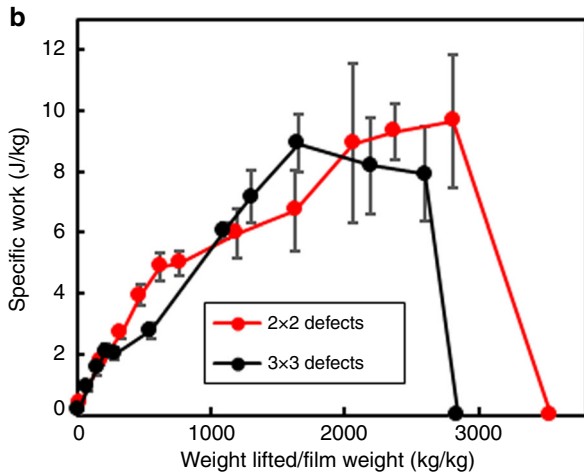

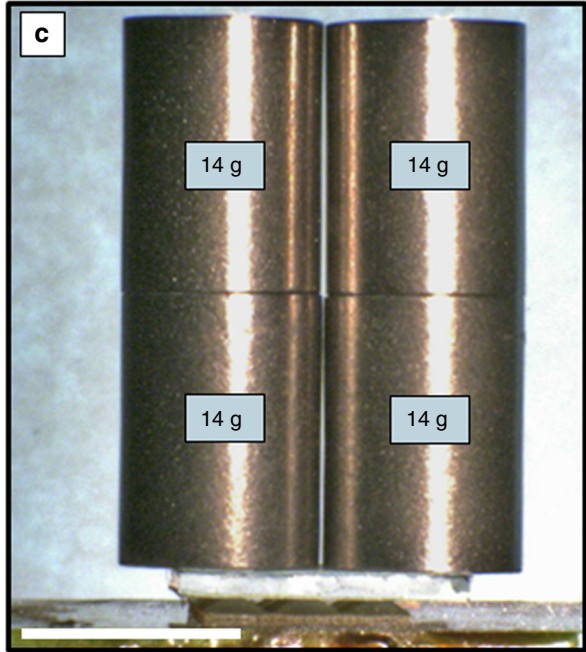

**Fig. 4** Force enhancement. **a** Stroke as a function of load for 1 × 1 cm films patterned with 2 × 2 or 3 × 3 +1 defect arrays. **b** Specific work of the 2 × 2 and 3 × 3 arrays. Errors bars represent tilt of the lifted weight and range of three experiments. **c** Illustration of the deformation of a 1 × 1 cm, four layer LCE laminate (26 mg) lifting over 56 g of load over 0.4 mm. Scale bar is 1 cm

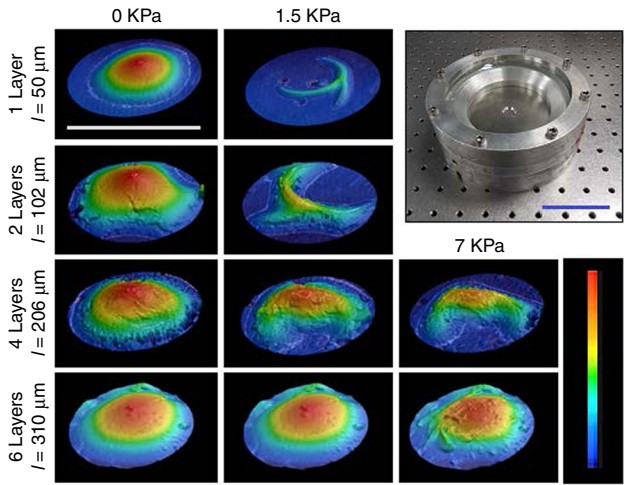

**Fig. 5** Shape deformation under positive pressure. LCE films and laminates (10 mm diameter) patterned with +1 topological defects subsumed in the center were heated to 100 °C in a pressure chamber (top right). Air pressure is applied on one side of the films, and the deformation measured via optical profilometry. The horizontal scale bar is 1 cm, while the height scale is 2.4 mm. The scale bar of the pressure chamber is 5 cm

**Materials characterization**. Phase transitions, birefringence, and film quality were measured via polarized optical microscopy (POM) (Nikon) in transmission mode, and the temperature was controlled by a Mettler Toledo HS82 heat stage. Contraction and areal change of homogenous planar films, floating on silicone oil and 5 μm glass spacers, as a function of temperature was also determined using POM. Differential scanning calorimetry (DSC) (TA Instrument Q1000) was performed under nitrogen from −40 °C to 120 °C for monomer mixtures and 0 °C to 300 °C for cured films in hermetically sealed pans at 2 °C/min. The nematic transition determined from the peak of the heat flux trace on second cooling, and the glass transition was determined from the peak of the derivative of the heat flux trace. Shape of actuated samples was measured through structured-illumination optical profilometry (Keyence VR-3200).

To measure lifting force and stroke of the actuation, 1 × 1 cm samples were placed on a resistive heater, loaded with weight, and heated to 180 °C. The samples were loaded with successively heavier tungsten weights and/or glass slides after each test, and the height was measured by a CCD camera at the plane of the films. A ruler was always imaged frame and plane to calibrate the distance, and the film displacement was measured using ImageJ. All tests were performed by loading the sample and then heating.

Actuation under pressure was measured using a homebuilt pressure chamber, and the resultant shape change monitored in situ via optical profilometry. The sample was placed in the central hole, with a series of small holes cut out under the sample to allow the back to be exposed to ambient pressure. An exploded drawing is displayed in Supplementary Figure 6. The chamber was heated to 100 °C using resistive heating silicone elements, and allowed to equilibrate for 30 min at 100 °C. The chamber was then pressurized until the films collapsed. The shape change was monitored every 0.2 psi.

**Data availability**. The authors declare that all data supporting the findings of this study are available within the paper and its Supplementary Information.

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

## Acknowledgements

This work was financially supported by the Air Force Office of Scientific Research and the Air Force Research Laboratory.

## Author Contributions

Initial research directions were identified by T.G., R.V.B., G.W.B., and T.J.W. Experimental examinations were undertaken by T.G. with assistance from M.J.S., B.A.K., and A.D.A. All authors contributed to the writing of the manuscript.

## Additional information

**Competing interests:** The authors declare no competing interests.

