## [Peer Review File · Nature Communications]

Reviewers' comments:

Reviewer #1 (Remarks to the Author):

It is true that it is very difficult to prepare thick LCE films with well aligned mesogens because of the limitation of the alignment layer in anchoring energy to make the mesogens aligned throughout the entire cells. This paper deals with a preparation method to obtain thick LCE films with the same composition and complex alignment of mesogens with the aid of photoalignment, which sounds a clever and elegant method. The resultant LCE films can carry objects 2,500 times heavier than themselves. I recommend publication of this paper in Nature Communications after the authors respond to the following comments satisfactorily.

1) Figure 2: There are two (d)s in the figure caption. The first (d) should be (c).

2) Methods: 'Dynamic scanning calorimetry (DSC)'

DSC should be 'Differential scanning calorimetry'.

3) Reference section is poorly written. In some cases details are not indicated:

ref. 25: no details; ref. 34: no journal name.

Reviewer #2 (Remarks to the Author):

Summary

A reversible actuator is claimed that comprises of laminated layers of photoaligned LC elastomers. The reversible response, in the form of out-of-plane lifting action, can achieve large stroke (millimeters) against considerable loads (a few hundred mN) from a small 1cm^2 2×2 array of features that are only a few hundred microns thick.

As cited, reversible out-of-plane actuation has already been demonstrated (Ware et al, Science, 982, 2015). The novelty here is a method to stack photopatterned layers to achieve thicker samples than can perform more work. Interestingly, as theory predicts, the maximum feature height does not change as more layers are integrated into the device.

Overall, it is a compelling story, well written, and the work is very convincing. If the manuscript must be shortened, then consider reducing/ removing the last section where actuation pushes against kPa of external pressure. This seems expected and not surprising to the least bit provided the earlier part of the manuscript.

I certainly feel the paper will indeed influence thinking in the field. It has influenced my thinking upon reading it!

Minor Issues and comments

- I personally do not favor statements like "the actuator is shown to displace a load more than 2500x heavier than its own weight" without mention or quantification of the stroke. T

- It would be interesting to compare the work capacity of these materials to entalpy of isotropization for the aligned and unaligned films.

- I would like to see some quantification of the adhesion between laminated films, or at least a comment about interlayer adhesion.

- Figure 1: I cannot tell haze from Figure 1b. The text reference to Figure 1c is incorrectly made to Figure 1d. "Contraction", the y-axis of Figure 1c is never defined. It looks like the response should be greater between 200-300 C; is this correct? The Figure legend is also incorrect; 1d and 1e are mixed up.

- I wonder if the 1cm^2 base dimension of the LC layers changes at all when actuation occurs. This should be mentioned or discussed because it may limit scalability of the phenomena. It may be worthwhile to report % height change-- it should be pretty high.

- The LCE laminates were placed in a pressure chamber where the back (or bottom) OF the film... (typo)

- reference 21 appears incomplete.

Reviewer #3 (Remarks to the Author):

The paper presents an approach toward the preparation of a soft actuator made of LCE, exhibiting an enhanced out-of-plane work capacity, when compared with other systems.

The film surfaces, obtained via photopolymerization at room temperature of a previously described nematic monomer mixture, is structured in a way to retain arrays of topological defect profiles (cones). On heating at high temperature, the film surfaces, which are flat at room temperature, are deformed to present an array of cones with a height of a few microns. By stacking several of those films in register, and gluing them using the same nematic monomers mixture, the resulting material can displace a load more than 2500 times its own weight.

Although the performances of this layered LCE actuator are impressive, there are drawbacks such as the very limited displacement (a few millimeters) and the high temperature needed to actuate the material (above 180°C).

Before being suitable for publication, several minor errors have to be corrected:

-page 2: preparation detailed in the supporting information: no information can be found in SI

-page 2 and figure 1: LC monomers which one is $n=3$, which one is $n=6$. Director profile is in figure 1d, not 1e

-page 3: Figure 1e and 1f: figure 1f does not exist.

Methods: it is not clear for me on how the films are stacked together; if each of the films are connected to a glass substrate, how do the stacked films can be made free from the glass substrate? Maybe as simple wrapping could help.

After the minor corrections made, the paper might be suitable for publication in Nature communications.

P.Keller

Author Response

Reviewer #1:

We are glad the reviewer found the work compelling and appreciate their thoughtful comments. Our response to the specific revision requests are detailed:

Comment: *1) Figure 2: There are two (d)s in the figure caption. The first (d) should be (c).*

Response: This has been corrected in the revised manuscript.

Comment: *2) Methods: 'Dynamic scanning calorimetry (DSC)' DSC should be 'Differential scanning calorimetry'.*

Response: We thank the reviewer for their careful reading and correcting this glaring mistake.

Comment: *3) Reference section is poorly written. In some cases details are not indicated: ref. 25: no details; ref. 34: no journal name.*

Response: Our original submission was plagued by a systemic issue with the referencing software (EndNote) we employed. We have corrected these references and corrected these and other issues in our citations.

Reviewer #2:

Again, we are glad to receive such positive comments from this reviewer.

Our response to specific revision requests now detailed:

Comment: *I personally do not favor statements like "the actuator is shown to displace a load more than 2500x heavier than its own weight" without mention or quantification of the stroke.*

Response: we have added the stroke length to the abstract.

Comment: *It would be interesting to compare the work capacity of these materials to enthalpy of isotropization for the aligned and unaligned films.*

Response: This is an interesting suggestion from the reviewer and not a subject we considered in this examination. Clearly, we would need to initiate a detailed study to capture the necessary data to initiate this correlation. We feel that this communication stands on its own, absent this data, and will pursue this experimentation and report their findings as merited in future studies.

Comment: *I would like to see some quantification of the adhesion between laminated films, or at least a comment about interlayer adhesion.*

Response: we have now inserted a statement, "Due to the consistency in the materials chemistry acting as both the LCE layer and adhesive, we observed no delamination in any of our experiments."

Comment: *Figure 1: I cannot tell haze from Figure 1b. The text reference to Figure 1c is incorrectly made to Figure 1d. "Contraction", the y-axis of Figure 1c is never defined. It looks like the response should be greater between 200-300 C; is this correct? The Figure legend is also incorrect; 1d and 1e are mixed up.*

Response: The figure caption has been revised, removing the statement "with low haze and scatter." Inferring from this comment, the reviewer is correct that we indeed do not report the haze values for these films. However, from the image, it is evident the samples are "optically clear".

Regarding Figure 1c reference, we infer that the term "shape change" is directing the reader to Figure 1d when in fact we are addressing Figure 1c. To clarify, we have replaced the term "shape change" with "contraction" throughout the text.

The Figure caption has been adjusted to correctly refer to the Figure panel d and e.

Comment: *I wonder if the 1cm^2 base dimension of the LC layers changes at all when actuation occurs. This should be mentioned or discussed because it may limit scalability of the phenomena. It may be worthwhile to report % height change-- it should be pretty high.*

Response: The base dimension of the LCE actuators does change upon heating. As the reviewer remarks, this limits the range of permissible shape change when anchored, as in Figure 5. We have now mentioned this in the main text.

We now also call out the percent change in height.

Comment: *The LCE laminates were placed in a pressure chamber where the back (or bottom) OF the film... (typo)*

Response: This typo has been corrected.

Comment: *reference 21 appears incomplete.*

Response: Again, our sincere apologies for issues with reference formatting. This reference (and others) have been corrected in this revision.

Reviewer #3:

Again, we thank the reviewer for their thoughtful commentary. We detail our revisions here:

Comment: *page 2: preparation detailed in the supporting information: no information can be found in SI*

Response: This reference has been removed.

Comment: *page 2 and figure 1: LC monomers which one is $n=3$, which one is $n=6$. Director profile is in figure 1d, not 1e*

Response: The monomer composition here is a mixture of RM257 ($n=3$) and RM82 ($n=6$). Thus, each sample contains both monomers. We have revised the methods section to clarify this.

We have corrected the labeling for Figure 1d and 1e.

Comment: *page 3: Figure 1e and 1f: figure 1f does not exist.*

Response: We have revised this to state “Figure 1d and 1e” (instead of Figure 1e and 1f).

Comment: *Methods: it is not clear for me on how the films are stacked together; if each of the films are connected to a glass substrate, how do the stacked films can be made free from the glass substrate? Maybe as simple drawing could help.*

Response: We now include an illustration in the Supporting Information to assist in the visualization of our methodology.

REVIEWERS' COMMENTS:

Reviewer #1 (Remarks to the Author):

The comments I made in the original manuscript have been satisfactorily addressed in the revised ms. I recommend publication of this paper in Nature Communications.

Reviewer #2 (Remarks to the Author):

The authors sufficiently addressed reviewer comments: I recommend publication.

Reviewer #3 (Remarks to the Author):

The revised version of this paper is now suitable for publication